# Swift Realisation of Wastewater-Based SARS-CoV-2 Surveillance for Aircraft and Airports: Challenges from Sampling to Variant Detection

**DOI:** 10.3390/microorganisms13081856

**Published:** 2025-08-08

**Authors:** Cristina J. Saravia, Kira Zachmann, Natalie Marquar, Ulrike Braun, Claus Gerhard Bannick, Timo Greiner, Peter Pütz, Susanne Lackner, Shelesh Agrawal

**Affiliations:** 1Wastewater Treatment Technology, Wastewater Disposal, German Environment Agency, 14195 Berlin, Germany; 2Chair of Water and Environmental Biotechnology, Institute IWAR, Department of Civil and Environmental Engineering Sciences, Technical University Darmstadt, 64287 Darmstadt, Germany; 3Wastewater Analysis, Monitoring Methods, German Environment Agency, 12307 Berlin, Germany; 4Department of Infectious Disease Epidemiology, Robert Koch Institute, 13353 Berlin, Germany

**Keywords:** SARS-CoV-2, wastewater surveillance, airport, aircraft, digital PCR, next-generation sequencing, wastewater sampling methods

## Abstract

International air traffic has contributed to the global spread of SARS-CoV-2 and its variants. In early 2023, wastewater-based epidemiology (WBE) has been implemented at airports as a surveillance tool to detect emerging variants at short notice. This study investigates the feasibility and challenges of applying WBE at Berlin Brandenburg (BER) Airport, including a rapid implementation of wastewater sampling and analysis under unprecedented circumstances. For this purpose, aircraft and airport wastewater was sampled over 13 weeks. Established sampling and analysis protocols for municipal wastewater treatment plants (WWTPs) had to be adapted to the specific conditions of the airport environment. SARS-CoV-2 RNA was quantified and sequenced, revealing SARS-CoV-2 mutations not previously observed in clinical surveillance data in Germany. Despite the logistical and methodological challenges, the study demonstrates that WBE can serve as an early warning system for pathogen introduction. However, our study also underscores the need for realistic timelines for the establishment and validation of WBE monitoring strategies in new contexts. Investments in the establishment of WBE systems, e.g., infrastructure, protocols, trained personnel, and a network of stakeholders at strategic nodes including airports, can act as an effective tool for pandemic preparedness and global health security.

## 1. Introduction

International travel facilitates the global spread of infectious diseases by transporting pathogens rapidly across borders. Due to the high volume of passengers and the long distances covered, air traffic is particularly impactful. In the case of coronavirus disease 2019 (COVID-19), a clear correlation emerged between international air travel volume and the spread of Severe Acute Respiratory Syndrome Coronavirus-2 (SARS-CoV-2) [1].

During the COVID-19 pandemic, wastewater-based epidemiology (WBE) has mainly been implemented for sewage surveillance in cities, providing crucial insights into community-level SARS-CoV-2 prevalence, while offering a cost-effective, complementary tool to clinical testing [2,3,4]. Adapting WBE to complex environments like airports and aircraft is not a straightforward extension of municipal wastewater surveillance systems—it requires tailored protocols, proactive planning, and administrative groundwork. These high-traffic locations offer strategic sampling opportunities but require specific adaptations rather than direct replication of municipal WBE protocols. Despite measures such as pre-testing and mandatory masks, ribonucleic acid (RNA) of SARS-CoV-2 was found in aircraft wastewater [5], highlighting the crucial role of WBE in tracking viral spread.

Although the rationale for WBE at airports is scientifically sound, its rushed deployment under pandemic pressure revealed major gaps in operational readiness. Unlike municipal wastewater surveillance systems with established infrastructure, airport surveillance efforts must be built from scratch—requiring navigation through regulatory, logistical, and technical hurdles. This highlights the urgent need for long-term, ready-to-deploy wastewater surveillance at high-risk travel hubs.

In December 2022/January 2023, there was concern for the emergence of new SARS-CoV-2 variants and their rapid spread to the European Union via international travel [6]. As the restrictions for international air travel were lifted, the authorities of the German Federal Ministry of Health recommended to monitor the situation via wastewater surveillance at airports.

By January 2023, wastewater-based monitoring of SARS-CoV-2 at municipal wastewater treatment plants (WWTPs) was well established globally. However, experience with wastewater surveillance specifically for airport and aircraft settings was generally limited, aside from a few initial studies and experiments conducted, such as those in the United Kingdom and Australia [5,7,8]. In response, the European Commission provided an ad hoc guidance on short notice [9]. Unlike municipal WWTPs, airports and aircraft lacked pre-existing sampling infrastructure, and implementing surveillance in these environments required developing effective sampling and analysis strategies due to the lack of prior experience. As this study illustrates, successful WBE at ports of entry depends not just on laboratory methods, but on systems-level preparedness. Surveillance at strategic travel hubs must be treated as core public health infrastructure, not temporary crisis interventions. This also reflects the European Commission’s plans to establish a cross-border early warning system by monitoring emerging infectious diseases in wastewater at so-called ‘Super Sites’—strategically selected locations, especially major transportation hubs (travel and trade) like international airports, seaports, and transit centres. Long-term investment is essential to ensure that sampling protocols, laboratory capacities, and stakeholder agreements are ready to deploy—not built in real-time during the next health crisis.

Establishing a sampling strategy for airports and aircraft presents several challenges. Samples must be representative of the surveillance scope (e.g., faecal matter from passengers of the sampled flight), technically feasible under time and resource constraints, and safe for personnel. Moreover, airport and aircraft access must comply with security regulations, requiring agreements with airport management. Various sampling locations can be targeted within aircraft wastewater systems and airport sewage networks. To obtain flight-specific samples, several studies recommend using a customised interception fitting placed between the aircraft sewage valve and the sanitary service truck hose. This enables direct collection of aircraft wastewater, avoiding mixing with wastewater from other sources [5,9,10,11,12]. Alternatively, samples can be taken from sanitary service trucks or wastewater collection points; however, this often results in composite samples representing multiple flights [7,8,9,10,13]. For the sampling of airport wastewater, composite samples using automated samplers are frequently recommended [7,13].

Regarding the quantification of SARS-CoV-2, the aircraft samples need to be concentrated first. In a few studies, a centrifugation step or a paper filter to separate the solids were used, followed by additional filtration steps [10,12,14]. After the extraction of viral nucleic acids, for the quantification of SARS-CoV-2 gene segments, mainly Real-Time qPCR (quantitative polymerase chain reaction) [12,14] or a combination with RT-ddPCR (reverse transcription droplet digital PCR) assays were conducted. Sequencing was performed to confirm positive aircraft samples and method specificity and for the detection of SARS-CoV-2 variants, such as Omicron [10]. In one study, SARS-CoV-2 gene fragments were detected near the detection limit in one of three aircraft samples [10]. Through the analysis of 198 commercial aircraft, Albastaki et al. [14] reported a percentage of positive signals of different SARS-CoV-2 genes of 13.6%.

Several studies have shown that the analysis of SARS-CoV-2 is feasible with a laboratory method, which is adapted to the potential low viral RNA load [15]. The detection of viruses in aircraft wastewater depends on the actual usage of toilets in the aircraft and the excretion rate of infected passengers [16,17,18,19]. Ahmed et al. [5] compared SARS-CoV-2 results from aircraft wastewater to clinical results from the respective passengers. They calculated a 50% probability to detect SARS-CoV-2 in wastewater if 2.8 infected passengers were on board and a 95% probability for 9.7 infected passengers [5].

Following the recommendations of the Federal Ministry of Health, we conducted a study to assess the feasibility of SARS-CoV-2 monitoring through the sampling and analysis of airport and aircraft wastewater at BER Airport, Germany. This study highlights the practical and institutional efforts required to implement wastewater surveillance at an international airport under tight timelines. It serves as a case study of what is needed for success, and why sustainable routines must be in place before a health crisis occurs. Here, we focused on the adaptation of suitable sampling and analysis methods. Furthermore, we investigated the added value of additional information on SARS-CoV-2 variants provided by wastewater samples from aircraft and airports. The data were compared to sequencing data from clinical samples.

## 2. Materials and Methods

### 2.1. SARS-CoV-2 Reference Data

The number of COVID-19 cases for calendar weeks 4 to 17 of 2023 and SARS-CoV-2 concentrations in municipal wastewater across German WWTPs were extracted from the open data platform of the Robert Koch-Institute (RKI, https://github.com/robert-koch-institut/SARS-CoV-2-Infektionen_in_Deutschland and https://github.com/robert-koch-institut/Abwassersurveillance_AMELAG/) (accessed on 29 August 2024).

Sequence information on SARS-CoV-2 variants from clinical samples in Germany were extracted from the German Electronic Sequencing Data Hub (DESH, now retired). The mutations recorded in the RKI DESH system are referred to as ‘clinical data’ mutations. The DESH dataset included all mutations detected in clinical samples in Germany where the frequency of at least one lineage reached 75% during the period from January 2023 to April 2023. Frameshift mutations were excluded from this analysis. Sequence data are now available on GitHub (https://github.com/robert-koch-institut/SARS-CoV-2-Sequenzdaten_aus_Deutschland, accessed on 15 June 2025).

### 2.2. Sampling Site and Period

In 2023, 23 million in- and outbound passengers travelled through BER Airport [20]. Furthermore, staff and visitors contribute to the wastewater generated at the airport, which is a mixture of wastewater from different sources, including lavatories, gastronomy, and cleaning. The airport has two separate sewers, one for wastewater from aircraft and one for the airport itself. The wastewater from the airport, hence all terminals, offices, and further buildings on the airport area is collected in a pump chamber. When reaching a volume threshold in the tank, the wastewater is transported in irregular intervals to the WWTP Waßmannsdorf via a pressure line.

In aircraft, the wastewater from lavatory sinks and galleys is usually directly evacuated from the aircraft via air valves. Hence, only the wastewater from toilets is collected in a tank. To reduce weight, toilets are flushed with a very small water volume (ca. 250 mL) in comparison to common toilets, which are flushed with several litres.

Aircraft wastewater is collected by sanitary service trucks directly upon landing. These tank trucks usually collect wastewater from several aircraft before transporting it to a central collection point. Using a valve at the bottom of the trucks, the wastewater is emptied into a tank through a funnel at floor level. A pressure line connects with the pressure line from the airport pump chamber and transports the wastewater to the WWTP Waßmannsdorf in irregular intervals.

Between late January and late April 2023, sampling took place 13 times at BER Airport on the same weekday between 05:30 a.m. and 06:30 a.m. Total traffic figures in the period were approx. 50,000 aircraft movements, which corresponded to approx. 6.3 million passengers or 418 flight movements and approx. 53,000 persons per day. Samples were taken from the first incoming flight of the day, a direct connection from overseas. The aircraft had a capacity of approx. 300 passengers. Based on the feedback from the sanitary service provider at BER Airport, the number of passengers ranged between 158 and 235 in the testing period, which resulted in an average of 198 passengers per flight.

### 2.3. Sampling of Aircraft and Airport Wastewater

Due to the technical conditions of the airport’s sewer system, the sampling method had to be modified. Typically, routine WWTP sampling methods employ automatic samplers to take 24 h composite samples. To sample the airport wastewater (hereinafter ‘airport sample’), grab samples were taken from the pump chamber using a 1 L dipping bottle, which was lowered into the chamber. Due to the position of the manhole and the limited access, sampling was only feasible at one spot. It was not possible to stir or homogenise the several cubic metres of wastewater in the chamber before taking the samples. The immersion depth of the bottle was approx. 0.5 m for each sample, disregarding different water levels in the pump chamber. For each sample, six individual samples of 1 L were taken and emptied into a bucket. After the homogenisation of the composite sample with a ladle, two subsamples of 1 L each were filled into polyethylene (PE) bottles.

The sanitary service provider supplied a pre-cleaned service truck to empty the aircraft’s tank. Afterwards, it was brought to the wastewater collection spot. There, grab samples were taken from the sanitary service truck tank (hereinafter ‘aircraft sample’). A 500 mL ladle was used to sample the tank through two manholes on top of the tank. For each individual sample, the tank’s content was first stirred with the ladle and then the sample was taken and emptied into a bucket. This way, eight samples were taken at each of the two manholes, resulting in a total volume of approx. 6 L. After stirring the composite sample with the ladle, two subsamples were filled into 1 L PE bottles. All samples were directly transported to the laboratories at 4 °C for subsequent PCR and sequencing analyses.

The described method for sampling aircraft was compared with a different method, in which samples were taken from the bottom valve of the sanitary service truck. A sampling scoop was used to catch the wastewater while the sanitary service operator opened and closed the valve. However, due to the relatively small wastewater volume in the tank (approx. 100 L) and the wastewater coming out splashing, it was challenging to collect enough wastewater for the aim of a 6 L composite sample.

To sample aircraft wastewater, appropriate protective equipment is needed. The person taking the sample had to climb onto the truck and it was almost impossible to work without contamination. Rubber boots, safety goggles (preferably face shields), disposable gloves, FFP-2 respiratory protection, and disposable splash suits were essential. All equipment had to be easy to disinfect and to be stowed away compactly in transport boxes, as the material needed to pass security checks.

### 2.4. Wastewater Parameters

Directly upon arrival to the laboratory, the pH value, electrical conductivity, and the total solids (TSs) of the samples were determined. The pH value and electrical conductivity were measured with respective sensors (Mettler Toledo FiveGo F2/3 with InLabExpert Go-ISM and InLab 738, Mettler Toledo, Columbus, OH, USA) directly in the PE bottles. To assess the TS, 50 mL of each sample were dried in a drying cabinet at 40 °C and the remaining mass was determined. Initial attempts to determine the total suspended solids (TSSs) by filtration, following the European Standard EN 872:2005 [21], were not successful due to the high viscosity of the aircraft wastewater.

### 2.5. Quantification of SARS-CoV-2 RNA

After analysing the wastewater parameters, the concentration of SARS-CoV-2 genes N2 and E were determined. Following the established methods for municipal WWTPs, the airport samples were treated according to the method established for municipal WWTPs. After homogenisation (10 min, 10 rpm), a centrifugation step (30 min, 4000 g, 18 °C) was carried out to separate solids, before 40 mL of the aqueous phase were concentrated into 1 mL using vacuum-based filtration. The applied protocol by Promega (Wizard Enviro Total Nucleic Acid Kit, Madison, WI, USA) combined sample concentration and RNA extraction via various washing steps with different buffers using silica membrane columns according to the manufacturer’s instructions. To minimise the duration of sample treatment, the airport samples were equally divided between two filter columns and combined before extraction. RNA was taken up in 40 µL or 80 µL of nuclease-free water and was aliquoted for subsequent analysis (storage at −80 °C). SARS-CoV-2 RNA was quantified by using ddPCR (Bio-Rad, Hercules, CA, USA). The RNA (9.9 µL per reaction) was transcribed into cDNA (complementary deoxyribonucleic acid) according to the manufacturer’s protocol of the PREvalence SARS-CoV-2 Wastewater Quantification Kit. This step is directly linked to the subsequent PCR in a ‘one-step’ procedure so that the target genes N2 and E were amplified (see Appendix A). A negative control was carried out in each PCR run. After the PCR, the droplets were stabilised for 45 min in the refrigerator, then analysed in the Bio-Rad droplet reader QX200 and quantified using the QX Manager Software 1.2 Standard Edition.

The aircraft samples were hardly diluted compared to the samples from the airport and WWTPs, since only small amounts of water are used when flushing the aircraft toilet. As a result, the filter units were clogged already during concentration and could not be processed to the usual extent (total sample volume). The attempt to enhance the sample preparation (e.g., different filter sizes, increased sample volume, additional centrifugation, other concentration/extraction kits, etc.) did not show an improvement and the samples contained hardly any gene copies of N2 and E of SARS-CoV-2. Due to the mentioned hurdles, the ddPCR data of the aircraft samples could not be evaluated. 

### 2.6. Sequencing of SARS-CoV-2

For sequencing analyses, the airport wastewater samples were first subjected to a preliminary sedimentation step for 10 min to allow settling and removal of solids, which could otherwise cause clogging of the filter membranes during ultrafiltration. Subsequently, 100 mL of the airport sample were concentrated by ultrafiltration using the centrifuge filter 10 kDa Centricon Plus-70 (Merck, Darmstadt, Germany). Due to the high content of paper and faecal material in the aircraft samples, the sedimentation step was modified by centrifuging the samples for 3 min at 1000 g, followed by concentration of a smaller volume of 50 mL supernatant. The aircraft samples were prepared as technical duplicates.

SARS-CoV-2 RNA was extracted from the concentrate using the MagMAX Microbiome Ultra Kit (Thermofisher Scientific, Waltham, MA, USA) and the King Fisher Duo Prime according to the manufacturer’s protocol. The extracted sample was then purified using Quick Clean Spin Filters (MP Biomedicals, Santa Ana, CA, USA).

To assess the quality of the processed wastewater samples and analyse the SARS-CoV-2 concentration, in contrast to the SARS-CoV-2 quantification method described in 2.5, the digital PCR (dPCR) was performed using the QuantStudio Absolute Q Digital PCR System (Thermo Fisher) and two different kits according to the manufacturer’s instructions: the Absolute Q dPCR SARS-CoV-2 Wastewater Surveillance Kit and the QuantStudio Absolute Q M16 Digital PCR Kit.

cDNA synthesis was performed using the SuperScript VILO cDNA synthesis kit according to the manufacturer’s protocol (Thermo Fisher Scientific) prior to sequencing on an Ion S5 sequencer (Thermo Fisher Scientific) utilising the IonTorrentTM technology. Library preparation was conducted with the Ion AmpliSeq SARS-CoV-2 Research Panel, comprising 237 primer pairs with amplicon lengths of 125–275 base pairs, covering nearly the entire SARS-CoV-2 genome. Libraries were amplified and sequenced on an Ion Torrent 550 chip, accommodating up to 8 samples per chip, yielding over 10 million reads per sample. All samples were sequenced in technical duplicates.

The detected genome sequences were compared with the SARS-CoV-2 reference sequence from Wuhan (Wuhan-Hu-1-NC_045512/MN908947.3), whereby the mutations were determined from the deviations. To assess the quality of the sequencing and the impact of the wastewater composition on the data, the SARS_CoV-2_coverageAnalysis plugin (v5.16.0.4) in Torrent Suite software (v5.18.1) was used. Variant annotation and analysis of base substitution effects were performed using the SARS_CoV-2_annotateSnpEff plugin (v5.16.0.5).

The Freyja tool was used to characterise the SARS-CoV-2 variants [22,23]. Freyja is a software tool developed to estimate the relative frequency of different SARS-CoV-2 lineages in mixed samples from a sequencing dataset aligned to the reference genome Wuhan-1. By analysing the frequency of single-nucleotide variants (SNVs) and the sequence depth of each SNV at specific positions in the genome, Freyja provides an estimate of the actual lineage frequencies in the sample [22]. The SARS-CoV-2 sublineages are shown as parent lineages according to the pangolin classification [24].

## 3. Results

### 3.1. SARS-CoV-2 Reference Data

From a peak in December, the number of clinical COVID-19 cases in Germany declined in the first three calendar weeks of 2023. However, cases began to increase again, reaching a new maximum of 113,452 cases in calendar week 9, which equals an incidence of 147 cases per 100,000 inhabitants. The numbers decreased again, steeply at first then slowly until the end of the observed time frame. Cases in Berlin declined until calendar week 3, remained fairly stable until calendar week 9, decreased again at first and then increased steeply for one week, after which they decreased slowly—with an exception between calendar week 15 and 16—until the end of the observed time frame (Figure 1 top). The SARS-CoV-2 viral load in wastewater, as monitored through the German WBE project, was similar to the trend observed in the notification data in Germany. Here, an increase was recorded up to calendar week 10, then a decrease up to calendar week 14. Over the next three weeks, the viral load remained relatively stable. The SARS-CoV-2 viral load in wastewater in Berlin did not fluctuate much during the observed time frame.

### 3.2. Adaptation of Workflows

Sampling and analysing wastewater from airport and aircraft is not a routine task, and the development and adaptation of sampling, quantification, and sequencing protocols was a process that took several weeks (see Figure 2 for a detailed timetable). The developed methods had to provide results that were representative of the prevalence of SARS-CoV-2 among aircraft passengers and airport users. Furthermore, routine monitoring needs to be actionable, reducing required time and effort to a reasonable degree.

A major challenge when establishing airport and aircraft sampling was the integration into the airport’s operating processes and security regulations. There was no protocol for wastewater sampling at BER Airport and therefore no internal agreements. Hence, the communication with several different teams had to be established first, such as the sanitary service provider, the wastewater staff, and the security management. The airport sewage system and circumstances had to be analysed to identify suitable sampling spots. This included on-site visits together with personnel from the mentioned teams. Access to the runway was needed to sample aircraft. In the case of BER Airport, the number of accesses as a visitor is limited to twelve visits per year and person. Hence, individual access passes had to be requested. This was linked to a security training and exam as well as a personal security check by the aviation security authority. Approximately eight weeks had to be scheduled to complete the process. However, sampling could be carried out when accompanied by authorised personnel beforehand. This was possible two weeks after the first request, due to the active cooperation of the airport administration.

When sampling airport wastewater, it is crucial to obtain a sample which is representative for the airport users in a specific timespan. However, it was not possible to homogenise the wastewater in the pump chamber and to determine from which exact period the wastewater originated.

For taking aircraft samples, samples must be representative of the virus fragments shed by the passengers of one or multiple flights. Furthermore, taking samples must be safe for employees. At BER Airport, in consultation with the sanitary service provider, the most viable option was by sampling the sanitary service trucks. To avoid the mixing with wastewater of another aircraft, a clean truck was provided for each sampling, although this was an additional task to the normal routine. Two different methods were compared to sample the sanitary service truck as described in Section 2.3. When sampling through the manhole of the truck, it was easier to collect enough sample volume. Furthermore, the possibility to stir the wastewater before sampling helped to collect a homogenised sample. In consequence, sampling through the manhole of the truck was preferred and established approximately six weeks after the first sampling. The protocols already established for the quantification of SARS-CoV-2 via ddPCR for municipal WWTPs could be applied to the airport samples within the first few weeks, whereby the samples were divided between two filter columns to reduce the processing time. The aircraft samples, which were tested between weeks 4 to 8, were not processed further after this period due to problems with sample preparation.

Due to the higher loads of solids, paper, and faecal matter in airport and aircraft wastewater compared to municipal wastewater, the sequencing workflow was adapted. This took two and six weeks, respectively. Smaller sample volumes were used for both airport and aircraft wastewater. A sedimentation step was introduced to prevent filter clogging in airport samples, while a brief centrifugation prior to ultrafiltration in aircraft samples reduced processing time (<12 h ultrafiltration time). Dilution tests confirmed that undiluted extracts yielded the best sequencing results without inhibitory effects. All samples were sequenced in technical duplicates to ensure data reliability, particularly critical for aircraft-derived material.

### 3.3. Wastewater Composition

The basic composition of the wastewater samples was analysed by determining the parameters TS (Figure 3), electrical conductivity, and pH value (see Appendix A, respectively). The airport samples showed significant variation between the sampling days for TS and electrical conductivity. The TS ranged from 1.2 g/L to 13.6 g/L (median: 3.0 g/L) and the electrical conductivity ranged from 1596 µS/cm to 4752 µS/cm (median: 2139 µS/cm). The aircraft samples were highly viscous and contained large amounts of paper waste, making the determination of the water parameters challenging. The TS and electrical conductivity showed less variation compared to the airport samples. The aircraft samples contained a higher amount of TS than the airport samples, ranging between 22.4 g/L and 35.7 g/L (median: 29.0 g/L). The electrical conductivity of the aircraft samples was lower than that of the airport samples in all weeks except one and showed less variation, ranging from 1187 µS/cm to 2487 µS/cm (median: 1537 µS/cm).

### 3.4. SARS-CoV-2 Quantification

The airport and aircraft samples were measured with the routine workflow for municipal WWTPs to quantify the SARS-CoV-2 genes E and N2. Compared to the raw wastewater samples from WWTPs, the aircraft samples had a very heterogeneous composition due to the proportion of recognisable coarse pieces and toilet paper, also reflected by the higher proportion of solids (Figure 3), could not be processed and are not displayed.

As shown in Figure 4, the SARS-CoV-2 concentration in the airport samples rose from calendar week 6 with below 100 gc/mL E and N2 to week 9 with 1142 gc/mL E and 1287 gc/mL N2, before declining again over the next two weeks. Elevated concentrations of the virus were also detected in calendar week 12 with 849 gc/mL E and 1053 gc/mL N2. The concentrations of SARS-CoV-2 genes E and N2 were similar, with E being about 24% smaller on average.

### 3.5. Sequencing

The influence of sample composition on sequencing quality was particularly evident in aircraft samples. The percentage of reads on target (Figure 5A) was markedly lower in aircraft samples, with a median of just 3.7% (range: 1.0–99.2%) compared with a median of 94.9% in airport samples. In line with this, the target base coverage at 100× (Figure 5B) varied considerably in aircraft samples, ranging from 0.1% to 99.8%, whereas all sequenced airport samples achieved high coverage between 68.9% and 99.9%. The average base coverage depth (Figure 5C) showed a similar pattern: aircraft samples had a median depth of 1095 reads per base, while airport samples reached a median of 21,578. For samples with low coverage, as well as a low proportion of reads on target and low sequencing depth, dPCR analysis (Figure 5D) also indicated either no or only low concentrations of SARS-CoV-2.

Figure 6 (Top) presents the prevalence of SARS-CoV-2 variants detected in wastewater samples collected from both the airport and aircraft using the Freyja tool. Variants with a prevalence of less than 1% per sample are summarised under the category ‘Others’. Furthermore, the total number of distinct SARS-CoV-2 variants detected using the Freyja tool is displayed for each sample, considering both sequenced duplicates, showing if they are categorised as either ‘Others’ (prevalence lower than 1%) or ‘Non-Others’ (prevalence higher than 1%).

From calendar weeks 4 to 8, a high diversity in the SARS-CoV-2 variants was detected in the wastewater at BER Airport. During this period, the variants BA.5, BQ.1.1*, BE.1*, BM.1*, CH.1.1*, BN.1*, BY.1*, XBB.1.5*, and XBB.1.9* were present. From calendar week 9 onwards, the omicron recombinant XBB* has been the dominant variant. The XBB* sublineages with the highest prevalences were XBB.1.9.1, XBB.1.5.2, XBB.1.5, and XBB.1.9.2. Compared to the SARS-CoV-2 variants detected in the airport wastewater, the sampling of the aircraft wastewater did not clearly show the dynamics regarding the shift in the dominant variants over time. In the samples from calendar weeks 4 to 9 BA.5, CJ.1.1, CW.1, BQ.1.27, BE.1.1.1, BQ.2, BA.2.75*, BA.2*, BR.2, XBB.1, XBB.1.5, XBB.1.9, and XBB.1.9.1 were the dominant variants. In calendar weeks 10 to 16, it was particularly noticeable that the variants XBB.1.9* with the sublineages XBB.1.9.1 and XBB.1.9.2 and XBB.1.5* with the sublineage XBB.1.5.2 were represented in high proportions in the six consecutive aircraft samples. This suggests that the other previously dominant variants were completely displaced. Nevertheless, BN.1* and BE.1* were again detected in high proportions in the sample in calendar week 17.

The analysis of SARS-CoV-2 variants in wastewater samples from airport and aircraft showed clear differences in total variant counts over the calendar weeks examined (see Figure 6, Bottom). Airport samples displayed high diversity, with 44 to 171 variants per sample, peaking in weeks 4 (171) and 7 (170). Aircraft samples had lower counts, ranging from 16 to 99 variants, with a peak in week 10 (99). A large proportion of the detected variants fell into the ‘Others’ category (prevalence < 1%). In airport samples, ‘Others’ consistently dominated, making up over 80 to 90% of total variants in several weeks, such as week 4 (154 of 171) and week 7 (151 of 170). Aircraft samples also showed a substantial share of ‘Others’, though generally lower, with a maximum of 81 of 99 (82%) in calendar week 10 (see Figure 6).

To further investigate the suitability of airport wastewater surveillance as a potential early detection system for SARS-CoV-2 mutations, the detection times of 2676 mutations identified in both clinical patient samples and wastewater samples from airport and aircraft were analysed and compared. Figure 7 presents an overview of the SARS-CoV-2 mutations detected earlier in wastewater surveillance from both airport and aircraft sources, and those detected earlier in clinical surveillance with patient samples (data from RKI DESH) or simultaneously in both surveillance types. The figure also illustrates the temporal dynamics of these mutations across the calendar weeks.

Of the mutations detected in wastewater and clinical surveillance, 2599 (97.1%) were detected at least one week earlier in airport wastewater (1967 mutations = 73.5%) and in aircraft wastewater (552 mutations = 20.6%), or simultaneously in both airport and aircraft wastewater (80 mutations = 3.0%) than in clinical surveillance data. Only 62 mutations (2.3%) were detected at least one week earlier in clinical sequencing data, and 15 mutations (0.6%) were detected simultaneously in both wastewater and clinical surveillance data in the same calendar week.

The analysis of temporal patterns of newly emerging SARS-CoV-2 mutations across calendar weeks 4 to 17 showed consistently high counts for mutations detected at least one calendar week earlier in airport wastewater, with notable peaks in weeks 6 (170), 7 (166), 8 (354), 9 (169), 13 (251), and 15 (175). Aircraft wastewater showed moderate levels in weeks 4 (61), 5 (51), 10 (180), 12 (75), 15 (67), and 16 (85), but very low or no detections in weeks 7 to 9 (1 to 4 mutations). Mutations detected simultaneously in both airport and aircraft wastewater were observed at lower levels, with small peaks in weeks 4 (41), 6 (6), 10 (1), 12 (6), 13 (5), 15 (7), and 16 (10). Clinical surveillance (>50,000 sequenced samples) showed overall low counts of earlier detected mutations than in wastewater, with sporadic detections between weeks 10 to 14 and a maximum of 26 mutations in week 10.

## 4. Discussion

In 2023, WBE at airports and on aircraft was not yet established in Germany, and workflows had to be developed. This required adapting international experiences [5,10,11] and iterative field testing, revealing major differences compared to municipal wastewater monitoring. However, the success of such efforts depended not only on laboratory protocols but also on addressing site-specific infrastructural, administrative, and technical challenges. These complexities underscore that WBE at high-security transportation hubs such as airports cannot be implemented as an emergency response without prior planning and investment. In the case of BER Airport, despite excellent conditions, it took 11 weeks after the first notification until routine monitoring was established.

Unlike municipal WWTPs, where sampling is a routine and integrated part of system design, airport and aircraft sewage systems often lack dedicated sampling infrastructure, as was the case for BER Airport. Gaining access to appropriate sampling points required detailed mapping of the airport’s wastewater system and coordination with multiple stakeholders. The lack of built-in sampling infrastructure made the collection of representative samples challenging, as grab samples were taken under operational constraints. Consequently, the representativeness of the airport samples is limited, as it was not possible to homogenise them and it is unclear what time period the sample covers. Furthermore, the sampling of aircraft is labour-intensive and requires extensive protective equipment to minimise hygiene risks. These findings align with earlier reports highlighting the unpredictability of sampling at airports [5,10,12]. At BER Airport, due to extensive administrative procedures regarding security and access regulations, the total preparation time for the first sampling was two weeks. These findings echo earlier assessments that emphasised the complex legal and logistical frameworks associated with surveillance at ports of entry [7,9,11]. For security and access regulations, international experience is not transferrable, e.g., Li et al. [25] emphasise the permission of airline companies to take aircraft samples, which was no requirement at BER Airport.

Aircraft wastewater is fundamentally different from municipal wastewater. It is highly concentrated, with a mean TS of 29.8 g/L, while the TSS of municipal WWTPs ranges between 0.1 and 0.6 g/L. Furthermore, it is subject to large temporal and compositional variability. Our findings confirm prior observations [10,16] that such samples are difficult to filter and may inhibit RNA extraction and amplification.

Only about 30 to 35% of passengers on long-haul flights use onboard toilets, and faecal shedding of SARS-CoV-2 occurs in 30 to 60% of infected individuals [16,17,18]. This means that the proportion of passengers shedding viral RNA into the wastewater can be low. As such, sensitive methods like digital PCR and sequencing are essential, and careful sample homogenisation is critical to avoid false negatives [10,16]. Our field evaluations found that sampling via manhole access on sanitary service trucks was most practical, allowing the homogenisation of contents. This sampling method contrasts with the experience from other studies, which took samples with a special interception device [8,11,12], sampled the sanitary service truck with a peristaltic pump [13], or took samples via the valve of the sanitary service truck [10].

In this study, ddPCR protocol previously used for monitoring SARS-CoV-2 in municipal sewage failed in one laboratory for the aircraft samples, while in another laboratory, dPCR on a different platform succeeded, and sequencing yielded high-quality data. Above all, it became clear that processing aircraft samples using filter-based concentration methods is time-consuming and that other protocols or pre-treatments would have to be tested to generate results as quickly as possible. This reinforces the importance of platform-specific performance testing and matrix-specific adaptation to address challenges such as high solids content and inhibitory substances [13,18]. These findings illustrate the critical need to allow sufficient time to validate protocols for each wastewater matrix, as methods optimised for municipal WBE are not necessarily transferable to aircraft or airport wastewater [5,8,12,13,18,19].

SARS-CoV-2 genome sequencing revealed circulating variants and additional mutations, some of which were not captured in concurrent clinical datasets. Notably, this was achieved despite analysing just 26 wastewater samples, compared with over 50,000 sequenced clinical patient samples. This confirms that airport and aircraft wastewater can function as complementary surveillance tools for detecting variant importation, consistent with international reports from Australia [5], the United States [11], and Dubai [14]. In France, Germany, the United Arab Emirates, and Australia, WBE at airports detected the Omicron variant earlier than or simultaneously with clinical surveillance, validating its use as a border-level early warning system [5,8,12]. While all aircraft samples showed SARS-CoV-2 RNA, airport samples tended to produce higher sequencing quality due to larger volumes, better mixing, more stable flow, and lower inhibitor concentrations. Notably, variant calling tools such as Freyja enabled detection of minor lineages within complex sample mixtures [22,23]. In Berlin, the maximum number of potentially sampled individuals per aircraft wastewater collection ranged from 158 to 235 passengers. Nevertheless, SARS-CoV-2 RNA was detected in every sample, demonstrating the feasibility of sampling individual aircraft toilets for genomic wastewater surveillance. However, it is important to note that the variants identified in aircraft wastewater using the Freyja tool do not necessarily reflect the exact number of infected passengers. This discrepancy arises from the complexity of wastewater samples, low genome coverage, and the inherent limitations of bioinformatic analyses: the number of detected variants may exceed the number of infected passengers per aircraft. This is partly due to the assignment of short sequencing fragments to multiple reference variants, leading to the apparent detection of a broader range of variants [22,23]. Consequently, both the quality of the samples and the quality of the sequencing must be considered collectively when interpreting the results. Thus, airport terminal wastewater may be the more reliable matrix for continuous surveillance at travel hubs.

The challenges observed here mirror those likely to be encountered in other non-routine WBE settings—such as stadiums, border crossings, or mass gathering events [13,15]. These contexts share similar constraints: access limitations, variable wastewater quality, and tight timelines for result turnaround. Even under ideal conditions, optimising protocols for complex matrices like aircraft wastewater requires several weeks. This timeline is not compatible with urgent public health responses unless protocols and personnel are already in place. Delayed results can diminish the value of the data for outbreak response, travel policy, or containment efforts. Furthermore, the obtained results need to be interpreted carefully. The uncertainty in passenger movement and airport activities restrict their efficacy as a direct indicator for the epidemiological situation in countries from which the incoming aircraft originate. In addition to SARS-CoV-2, recent studies have demonstrated the detection of other respiratory viruses (e.g., influenza viruses, respiratory syncytial virus (RSV)) and enteric pathogens (e.g., adenovirus, norovirus, hepatitis A) in airport wastewater [13]. Therefore, the experience gained from this study is not only valuable for SARS-CoV-2 monitoring but may be transferred to further viruses or emerging pathogens in non-routine WBE settings. This expands the potential of airport WBE as a routine surveillance tool, not just a crisis-response mechanism.

In our study, we were able to show that sampling aircraft and airport wastewater is feasible, if all associated requirements are met. The case and viral load data from Berlin and Germany showed that SARS-CoV-2 infections were present during the observation period. Our data do not support nor exclude an impact of air traffic at BER Airport on local infection dynamics (Figure 1). By sampling ‘Super Sites’ such as large transportation hubs, e.g., airports, it is possible to generate an additional surveillance tool for pathogen detection (types, levels, and spread) across borders. Important are standardised protocols and timely reporting systems with functioning infrastructure, technology, and coordination as well as response capabilities for future pandemics.

## 5. Conclusions

This study demonstrates that WBE can be rapidly deployed at airports for SARS-CoV-2 quantification and variant detection. However, successful implementation requires careful consideration of site-specific circumstances, including tailored sampling strategies, matrix-adapted laboratory workflows, and close coordination across administrative and technical stakeholders. Laboratory protocols developed for municipal wastewater must often be significantly modified for complex matrices like aircraft wastewater, which can vary substantially from flight to flight. These challenges are not unique to airports; other high-risk or event-based settings—such as stadiums, border crossings, or mass gatherings—face similar logistical and methodological constraints. Furthermore, event-based sampling requires timely results to be actionable.

Despite these limitations, both PCR-based and sequencing-based methods were successfully applied at various airport sampling points. Sample characteristics, such as viral RNA load and matrix composition, strongly influence data quality and must be considered when interpreting results. With appropriate adjustments and preparedness, airport WBE can serve as a powerful tool for early detection of emerging pathogens and their new variants at international points of entry. To improve readiness for future threats, sampling strategies should be tailored to the intended resolution—flight-level, composite, or terminal—and WBE systems should be treated as long-term public health infrastructure, not temporary emergency tools.

## Figures and Tables

**Figure 1 microorganisms-13-01856-f001:**
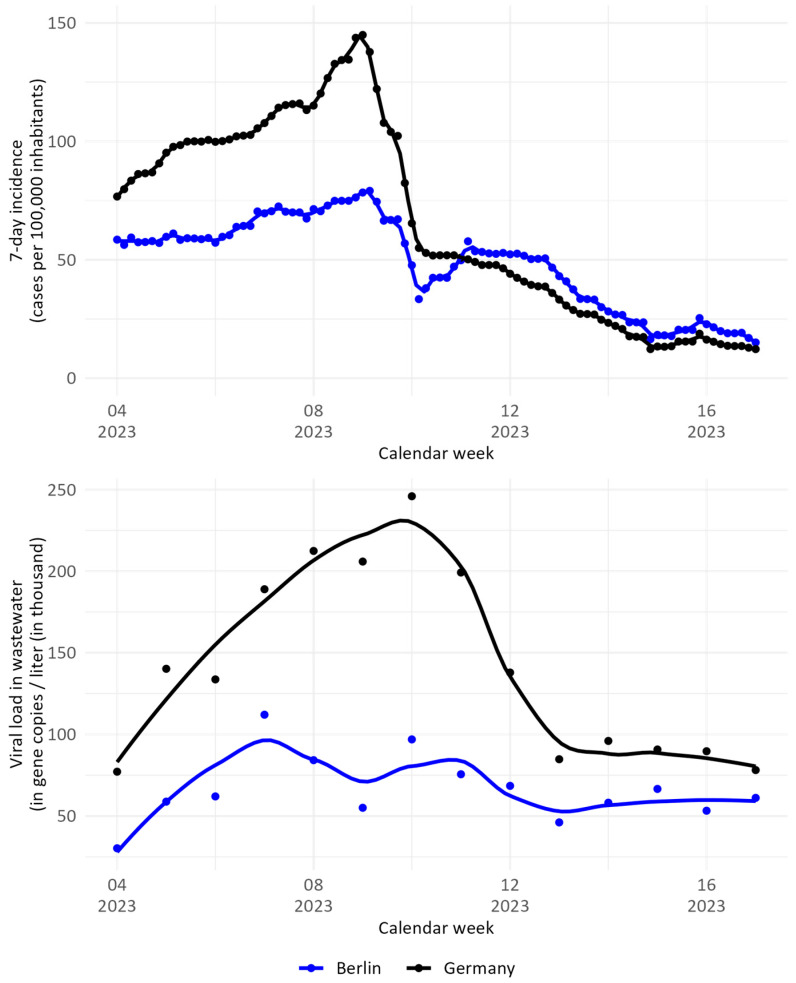
**Top**: COVID-19 incidence as reported to the German notification system database for Germany (black) and Berlin (blue). **Bottom**: SARS-CoV-2 viral load in wastewater in Germany (black) and Berlin (blue).

**Figure 2 microorganisms-13-01856-f002:**
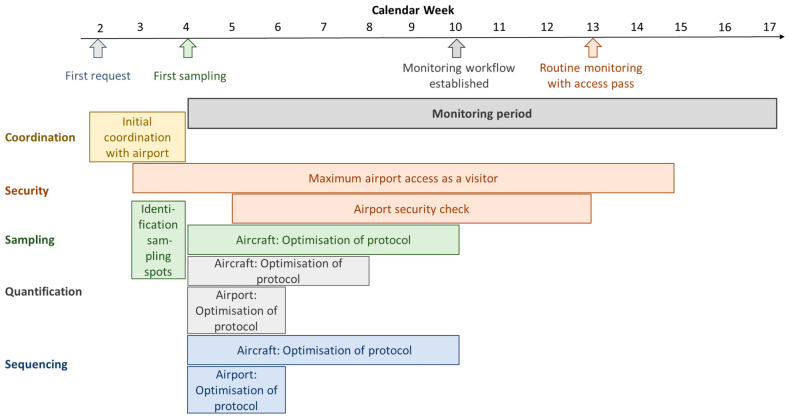
Timetable. Time periods that have been necessary to establish the SARS-CoV-2 monitoring at BER Airport, starting from the first notice.

**Figure 3 microorganisms-13-01856-f003:**
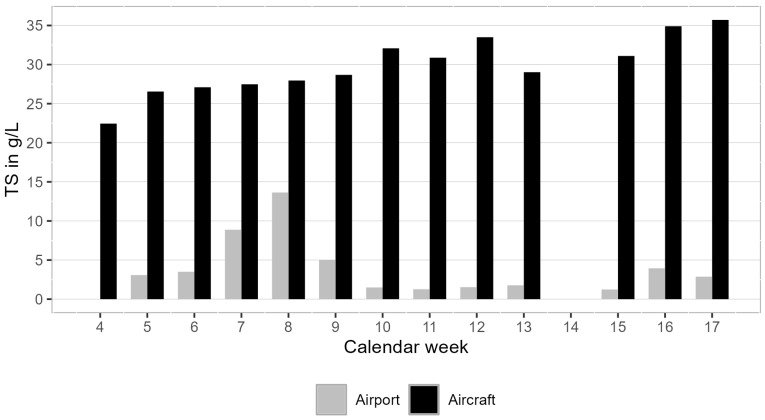
Total solids in the aircraft and airport samples for all sampling days (calendar week 14 is missed due to Easter holidays).

**Figure 4 microorganisms-13-01856-f004:**
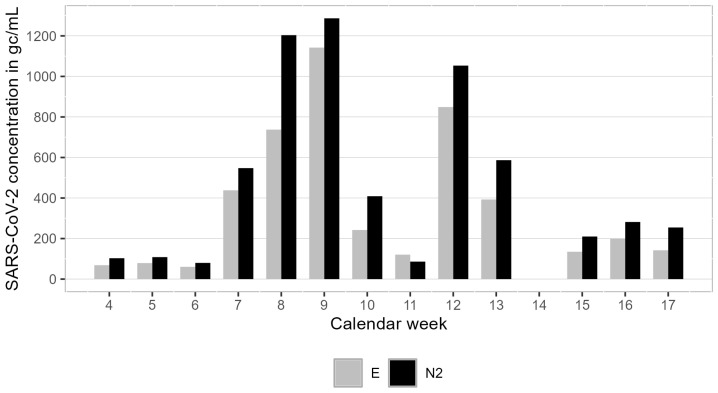
Gene copies of SARS-CoV-2 gene sequences E and N2 per mL wastewater derived from the airport samples for all sampling days (calendar week 14 is missed due to Easter holidays).

**Figure 5 microorganisms-13-01856-f005:**
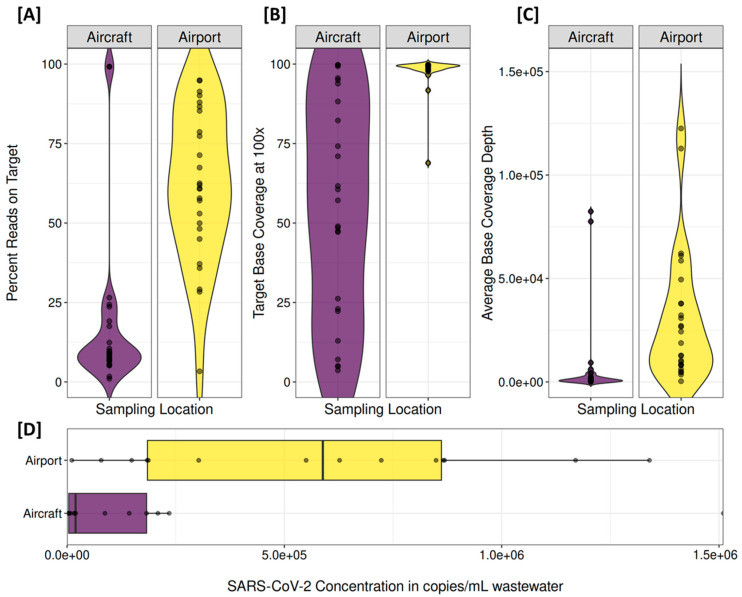
Impact of wastewater sample composition on sequencing quality. Sequencing metrics for aircraft and airport wastewater samples: (**A**) Percentage reads on target (mapped reads to the SARS-CoV-2 genome, indicating sequencing specificity). (**B**) Target base coverage at 100× (fraction of the genome covered at ≥100×, reflecting genome completeness). (**C**) Average base coverage depth (sequencing depth per base, representing the overall sequencing coverage). (**D**) SARS-CoV-2 concentration (measured by dPCR, providing an estimate of viral load). Sequencing duplicates are shown individually.

**Figure 6 microorganisms-13-01856-f006:**
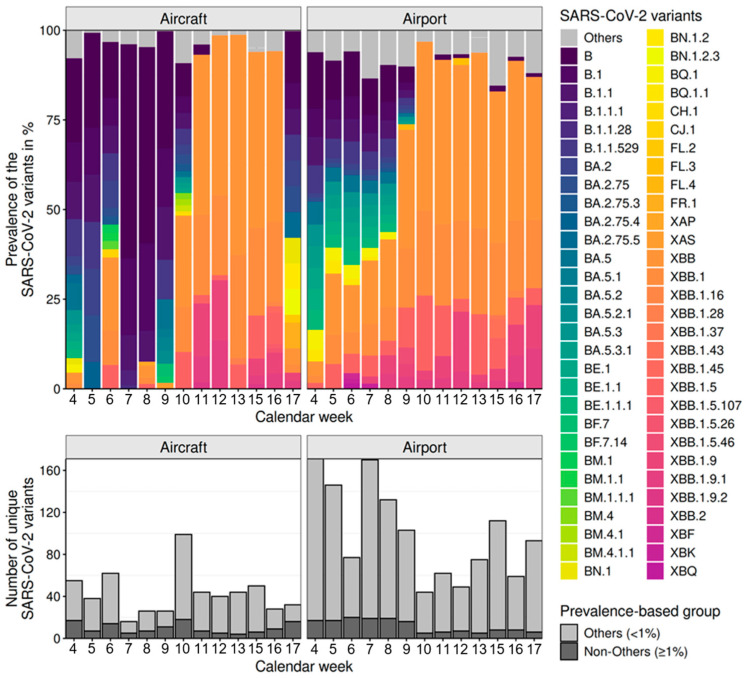
Overview of SARS-CoV-2 variants detected in airport and aircraft samples (analysed using Freyja). **Top**: Prevalence of SARS-CoV-2 variants; variants with a prevalence <1% are summarised as ‘Others’, and sublineages are grouped under their parent lineages according to the Pangolin classification. **Bottom**: Number of distinct SARS-CoV-2 variants per sample, based on combined sequencing duplicates, with the categories ‘Others’ (prevalence < 1%) and ‘Non-Others’ (prevalence ≥ 1%).

**Figure 7 microorganisms-13-01856-f007:**
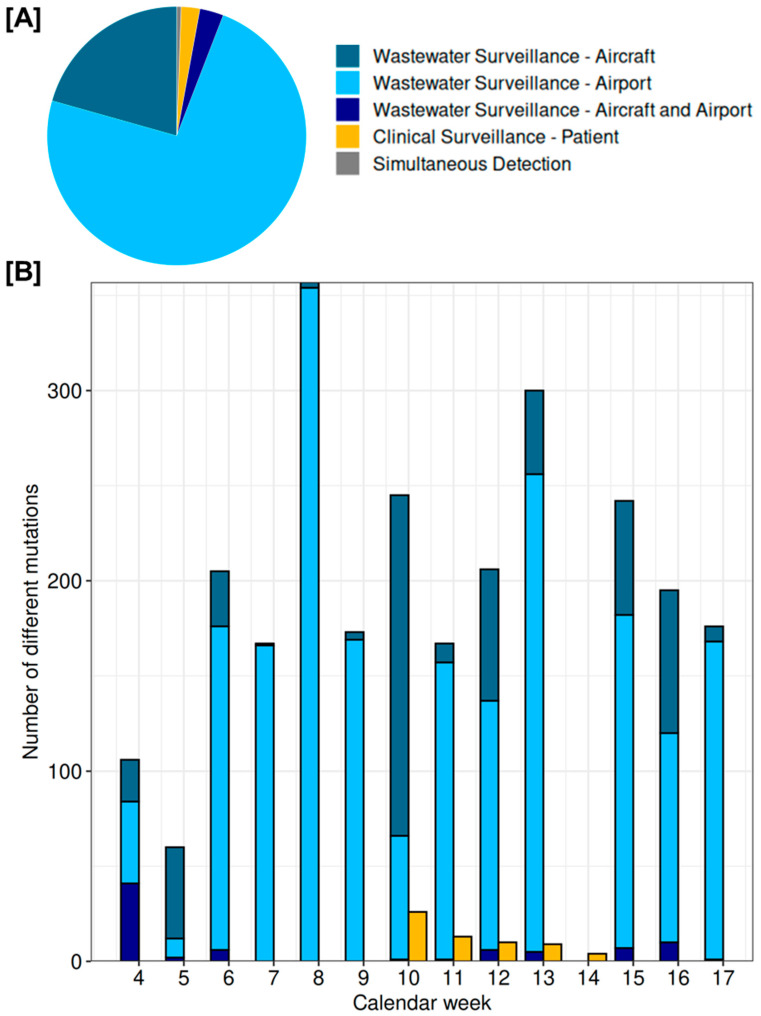
First detection of SARS-CoV-2 mutations across different surveillance strategies: wastewater from airport samples, wastewater from aircraft samples, and clinical samples from patients. (**A**) Share of mutations that could be detected in aircraft/airport/patient samples at least one calendar week earlier compared to the other surveillance systems (calendar weeks 4–17). Mutations detected in the same calendar week across multiple datasets are shown as ‘Simultaneous Detection’. (**B**) Total number of mutations first detected by the respective surveillance strategy each calendar week, with simultaneously detected mutations excluded.

## Data Availability

Data available in a publicly accessible repository. The processed SARS-CoV-2 ddPCR and dPCR data are available in the Appendix A. The raw sequencing data presented in this study are publicly available in the National Centre for Biotechnology Information (NCBI) Sequence Read Archive (SRA) under BioProject accession number PRJNA1283912 (Submission ID: SUB15345359). The FASTQ files for each sample can be found under the following accession numbers: SAMN49704293 to SAMN49704372.

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
