# Peer review of "Swift Realisation of Wastewater-Based SARS-CoV-2 Surveillance for Aircraft and Airports: Challenges from Sampling to Variant Detection"

_microorganisms, 2025, doi:10.3390/microorganisms13081856_

Round 1

Reviewer 1 Report

Comments and Suggestions for Authors

The manuscript by Marquar et al. details the initialization and results of an airport wastewater program set up to monitor SARS-CoV-2 epidemiology. The methods and results are well justified and integrate nicely with the growing body of wastewater base epidemiology knowledge. While the laboratory data answers the questions put forward by the research group, I feel there are some editorial changes required before publication. A brief analysis of off-target (non-SC2) sequencing reads would also be helpful.  They are as follows:

Major points:

  • Municipal wastewater analysis, particularly Section 3.1: The source for the municipal wastewater data (Fig 1, bottom) is not included in the methods. Is this city/state-collected data posted to a public repository? It doesn’t seem like the research group collected this for this study. Was it part of the RKI data? Please clarify the source of the data within the methods, and within the results.
  • Section 3.2: Please consider moving most parts of this section to the Methods and Discussion sections. I think a concise explanation of the set-up/sampling timeline, with a couple notable details, would be best. Study limitations/confounding issues can be moved to Discussion (some of which is already mentioned there). Unless data is presented or described (in main text or supplemental) preliminary testing/methods modifications can be described in Methods.
  • Section 3.5 and Figure 5: Please arrange figure sub-panels to match the order described in the text, or reorder text descriptions to match panel order.
  • A brief analysis of off-target (non-SC2) reads from the aircraft (and probably airport) sequencing reads would be interesting, since the airport/aircraft discrepancy is so dramatic. Presumably they will be mostly all human-mapping reads, with some gut flora, companion animal, and food-stuff related reads. Kraken would be a suitable software application, but other options are available. If warranted, the results could be another subpanel of Figure 5, otherwise a sentence in 3.5 would be fine (e.g., Most off-target reads in samples mapped to the human genome [mean XXX% +/- SD].)
  • Figure 7: Should there be another grouping: simultaneous aircraft/airport detection? I find it odd that all mutations would be found in either aircraft OR airport before the other.

Minor points:

Line 57: had been -> were

Line 160: A citation and/or brief description of the WWTP sampling/processing methodology is needed.

Line 207: The samples -> The aircraft samples (?). Also, this should probably be the start of a new paragraph.

Line 250: remove “the present”

Lines 259-261: This sentence is pretty long and a little confusing. Clarify, and break into two sentences.

Line 319: “tissues” -> paper waste (?), to clarify that this is not organic tissue.

Lines 438: Since it’s been a while since clinical mutations were discussed, remind the reader in the text of the source of the mutations (RKI DESH)

Author Response

The manuscript by Marquar et al. details the initialization and results of an airport wastewater program set up to monitor SARS-CoV-2 epidemiology. The methods and results are well justified and integrate nicely with the growing body of wastewater base epidemiology knowledge. While the laboratory data answers the questions put forward by the research group, I feel there are some editorial changes required before publication. A brief analysis of off-target (non-SC2) sequencing reads would also be helpful.  They are as follows: 

Major points: 

Comment 1: Municipal wastewater analysis, particularly Section 3.1: The source for the municipal wastewater data (Fig 1, bottom) is not included in the methods. Is this city/state-collected data posted to a public repository? It doesn’t seem like the research group collected this for this study. Was it part of the RKI data? Please clarify the source of the data within the methods, and within the results.  

Response 1: Thank you for pointing this out. Municipal wastewater data on viral loads have been collected through the German WBE project AMELAG, carried out by UBA & RKI. We included the source of the data in section 2.1, line 128. 

Comment 2: Section 3.2: Please consider moving most parts of this section to the Methods and Discussion sections. I think a concise explanation of the set-up/sampling timeline, with a couple notable details, would be best. Study limitations/confounding issues can be moved to Discussion (some of which is already mentioned there). Unless data is presented or described (in main text or supplemental) preliminary testing/methods modifications can be described in Methods. 

Response 2: Thank you for this valuable suggestion, we continuously discussed the distribution of our findings between the Methods, Results and Discussion sections. We think that the adaptation of the applied workflows and the corresponding processes are a key result of our study, thus are also part of the results section. However, we moved some of the mentioned parts to the Methods section, especially concerning sampling methods.

Comment 3: Section 3.5 and Figure 5: Please arrange figure sub-panels to match the order described in the text, or reorder text descriptions to match panel order.  

Response 3: We reorderd text descriptions to match panel order and specified the matching sub-panels by indicating the corresponding characters in section 3.5.  

Comment 4: A brief analysis of off-target (non-SC2) reads from the aircraft (and probably airport) sequencing reads would be interesting, since the airport/aircraft discrepancy is so dramatic. Presumably they will be mostly all human-mapping reads, with some gut flora, companion animal, and food-stuff related reads. Kraken would be a suitable software application, but other options are available. If warranted, the results could be another subpanel of Figure 5, otherwise a sentence in 3.5 would be fine (e.g., Most off-target reads in samples mapped to the human genome [mean XXX% +/- SD].) 

Response 4: We Thank you for this thoughtful suggestion. We’d like to clarify that our sequencing approach was based on the Ion AmpliSeq™ SARS-CoV-2 Research Panel (Thermo Fisher Scientific), which is a targeted amplicon-based method specifically designed for high-resolution detection of SARS-CoV-2 RNA. Unlike shotgun metagenomics, this method focuses on amplifying predefined viral genome regions and is not intended for broader taxonomic or microbiome profiling.

In this setup, any off-target reads—particularly in samples with little or no SARS-CoV-2 RNA—are typically the result of non-specific amplification noise, such as primer-dimer artifacts or low-level background (e.g., trace human RNA). These are well-characterized technical effects, and not a reliable reflection of the actual microbial or biological content of the sample. Studies using this panel (e.g., Alessandrini et al., 2020; Xiao et al., 2020) as well as the manufacturer’s own documentation consistently highlight this limitation.

While we understand the interest in exploring off-target reads, tools like Kraken are better suited to untargeted sequencing data, where read origin is more biologically meaningful. In our case, such an analysis would risk overinterpreting artifacts as signal, without adding real value to the study.

We hope this explanation helps clarify our reasoning and why we’ve chosen not to include off-target classification in the current analysis.

  • Alessandrini F, Caucci S, Onofri V, Melchionda F, Tagliabracci A, Bagnarelli P, Di Sante L, Turchi C, Menzo S. Evaluation of the Ion AmpliSeq SARS-CoV-2 Research Panel by Massive Parallel Sequencing. Genes (Basel). 2020 Aug 12;11(8):929. doi: 10.3390/genes11080929. PMID: 32806776; PMCID: PMC7463572.
  • Xiao, M., Liu, X., Ji, J. et al. Multiple approaches for massively parallel sequencing of SARS-CoV-2 genomes directly from clinical samples. Genome Med 12, 57 (2020). https://doi.org/10.1186/s13073-020-00751-4
  • Thermo Fisher Scientific(2021).
    Ion AmpliSeq™ SARS-CoV-2 Research Panel User Guide (Publication No. MAN0019260 Rev. B).
    https://assets.thermofisher.com/TFS-Assets/LSG/manuals/MAN0019260_Ion_AmpliSeq_SARS-CoV-2_Research_Panel_UG.pdf

Comment 5: Figure 7: Should there be another grouping: simultaneous aircraft/airport detection? I find it odd that all mutations would be found in either aircraft OR airport before the other. 

Response 5: We adapted the figure to include mutations which were detected simultaneously in aircraft and airport wastewater. 

Comment 6: Minor points: 

Response 6: We appreciate the reviewer’s minor points and adapted them in the text.  

Line 57: had been -> were  

Line 160: A citation and/or brief description of the WWTP sampling/processing methodology is needed.  

Line 207: The samples -> The aircraft samples (?). Also, this should probably be the start of a new paragraph.  

Line 250: remove “the present”  

Lines 259-261: This sentence is pretty long and a little confusing. Clarify, and break into two sentences.  

Line 319: “tissues” -> paper waste (?), to clarify that this is not organic tissue.  

Lines 438: Since it’s been a while since clinical mutations were discussed, remind the reader in the text of the source of the mutations (RKI DESH)  

Reviewer 2 Report

Comments and Suggestions for Authors

The study concept regarding WBE in airport is really interesting. The main methodological part of the manuscript does not actually refer to the molecular part, but in the sampling strategy and relevant procedures. Methodologically speaking, the manuscript is correct, supporting the outcomes. Thus, I do not have anything regarding methodology, but mostly I want to bring some points for thought and additions in the discussion.

Firstly, it would have been interesting to compare the same samples, if not all maybe a number, regarding the results with and without centrifugation step or a paper filter. This comparison would provide a nice guideline for the best proposed practice.

My main criticism is if the need for COVID surveillance is still needed. Recently, the problem of this pandemic has been to a great extent solved. Probably, this was not the situation when the authors designed the study, but this fact decreases the value of the findings. Maybe if the authors added a new part in the discussion subjecting COVID as a model for viruses,I believe would add benefits to the study.

The cost of this surveillance system is also an important aspect that should be discussed as well.

I believe that the sampling procedure described is the most important tool, adding value to the manuscript as a procedure, not necessarily for COVID

I have also some minor comments

Line 133: Any reference for this number?

It is not clear how many samples in total were collected. The procedure is described well in the last paragraph of 2.2, but the total number is not clear. If I am correct, it was 13 weeks?

Author Response

The study concept regarding WBE in airport is really interesting. The main methodological part of the manuscript does not actually refer to the molecular part, but in the sampling strategy and relevant procedures. Methodologically speaking, the manuscript is correct, supporting the outcomes. Thus, I do not have anything regarding methodology, but mostly I want to bring some points for thought and additions in the discussion. 

Comment 1: Firstly, it would have been interesting to compare the same samples, if not all maybe a number, regarding the results with and without centrifugation step or a paper filter. This comparison would provide a nice guideline for the best proposed practice. 

Response 1: Many thanks to the reviewer for this valuable suggestion. In this paper, the application of the workflow did not include the procurement and testing of various pre-treatment and preparation methods. The established procedures for municipal WWTPs were to be applied directly to the aircraft and airport samples. In contrast to other studies that compare different methods, the aim here was to test rapid transferability.  

Due to the lengthy concentration procedure, which included a very time-consuming centrifugation step, testing without initial centrifugation as pre-treatment was not considered. For future investigations of wastewater samples (not from municipal WWTPs with high solids content), further steps, such as the use of paper filters or similar (listed in lines 97/98), should be considered. This requires additional preparation time (see lines 566-573). 

Comment 2: My main criticism is if the need for COVID surveillance is still needed. Recently, the problem of this pandemic has been to a great extent solved. Probably, this was not the situation when the authors designed the study, but this fact decreases the value of the findings. Maybe if the authors added a new part in the discussion subjecting COVID as a model for viruses, I believe would add benefits to the study. 

Response 2: Our study primarily describes the challenges that arise from setting up a monitoring strategy for pathogens in airport and aircraft wastewater on short notice. The gained experience can be easily transferred and thus provides added value for the monitoring of further viruses. We added a sentence in the discussion, line 587, for clarification.  

Comment 3: The cost of this surveillance system is also an important aspect that should be discussed as well. 

Response 3: While the cost-efficiency of WBE systems is beyond the scope of this study it has been discussed extensively in literature. We included the cost-efficiency of WBE systems for the surveillance of virus infections in the introduction, line 44. 

Comment 4: I believe that the sampling procedure described is the most important tool, adding value to the manuscript as a procedure, not necessarily for COVID 

Response 4: We appreciate the reviewer’s comment and agree with the reviewer. While our study focused on SARS-CoV-2, we believe the methodology/ monitoring strategy described has broader relevance for the environmental surveillance of other pathogens or genetic markers in airport and aircraft wastewater.

Comment 5: I have also some minor comments 

Response 5: We appreciate the reviewer’s minor comments and adapted them in the text. 

Line 133: Any reference for this number?  

It is not clear how many samples in total were collected. The procedure is described well in the last paragraph of 2.2, but the total number is not clear. If I am correct, it was 13 weeks?  

Round 2

Reviewer 1 Report

Comments and Suggestions for Authors

Thank you for the modifications to the manuscript. I feel the paper has been sufficiently amended and approve publication in its current form.